# Digital Online Anaesthesia Patient Informed Consent before Elective Diagnostic Procedures or Surgery: Recent Practice in Children—An Exploratory ESAIC Survey (2021)

**DOI:** 10.3390/jcm11030502

**Published:** 2022-01-19

**Authors:** Claudia Neumann, Grigorij Schleifer, Nadine Strassberger-Nerschbach, Johannes Kamp, Gregor Massoth, Alexandra Görtzen-Patin, Dishalen Cudian, Markus Velten, Mark Coburn, Ehrenfried Schindler, Maria Wittmann

**Affiliations:** Department of Anaesthesiology and Intensive Care Medicine, University Hospital Bonn, Venusberg-Campus 1, 53127 Bonn, Germany; claudia.neumann@ukbonn.de (C.N.); grigorij.schleifer@ukbonn.de (G.S.); nadine.strassberger@ukbonn.de (N.S.-N.); johannes.kamp@ukbonn.de (J.K.); gregor.massoth@ukbonn.de (G.M.); alexandra.goertzen-patin@ukbonn.de (A.G.-P.); dishalen.cudian@ukbonn.de (D.C.); markus.velten@ukbonn.de (M.V.); mark.coburn@ukbonn.de (M.C.); ehrenfried.schindler@ukbonn.de (E.S.)

**Keywords:** telemedicine, digital informed consent, children, European practice

## Abstract

Background: One undisputed benefit of digital support is the possibility of contact reduction, which has become particularly important in the context of the COVID-19 pandemic. However, to the best of our knowledge, there is currently no study assessing the Europe-wide use of digital online pre-operative patient information or evaluation in the health sector. The aim of this study was to give an overview of the current status in Europe. Methods: A web-based questionnaire covering the informed consent process was sent to members of the European Society of Anaesthesia and Intensive Care Medicine (ESAIC) in 47 European countries (42,433 recipients/930 responses). Six questions related specifically to the practice in paediatrics. Results: A total of 70.2% of the respondents indicated that it was not possible to obtain informed consent via the Internet in a routine setting, and 67.3% expressed that they did not know whether it is in line with the legal regulations. In paediatric anaesthesia, the informed consent of only one parent was reported to be sufficient by 77.6% of the respondents for simple interventions and by 63.8% for complex interventions. Just over 50% of the respondents judged that proof of identity of the parents was necessary, but only 29.9% stated that they ask for it in clinical routine. In the current situation, 77.9% would favour informed consent in person, whereas 60.2% could imagine using online or telephone interviews as an alternative to a face-to-face meeting if regulations were changed. Only 18.7% participants reported a change in the regulations due to the current pandemic situation. Conclusion: Whether informed consent is obtained either online or on the telephone in the paediatric population varies widely across Europe and is not currently implemented as standard practice. For high-risk patients, such as the specific cohort of children with congenital heart defects, wider use of telemedicine might provide a benefit in the future in terms of reduced contact and reduced exposure to health risks through additional hospital stays.

## 1. Introduction

Particularly for seriously ill children, the risk of contracting nosocomial infections during a hospital stay is high. Children after cardiac surgery showed an overall nosocomial infection rate of 10.8% [1].

Even beyond the current pandemic situation, avoiding hospitalisation reduces the burden on seriously or chronically ill children. For example, an Australian study in 2019 showed that at-home antibiotic therapy for children with moderate to severe cellulitis had the same efficacy but fewer adverse effects compared to standard treatment in hospital [2].

Telemedicine is described as “the provision of concrete medical services overcoming spatial distances with the aid of modern information and communication technologies” [3]. This aid may provide many advantages, especially in severely ill children, who show a significant risk from hospital waiting-times and visits, particularly given the current need to limit social contacts during the pandemic. Numerous studies show that telemedicine care for patients is safe, effective, and cost-efficient [4]. The first publication regarding informed consent for anaesthesia using telemedicine dates back to 2004 and shows a high level of satisfaction among both anaesthetists and patients [5]. A study by Wool in 2016 evaluated the efficiency and reliability of telemedicine consultations for preoperative assessment of patients undergoing oral and maxillofacial surgery, including a six-year follow up period. Among other things, the authors concluded that in 98% of the cases, “Most patients received adequate medical and physical examination and were able to undergo surgery with anaesthesia as planned at the clinic appointment immediately after the telemedicine consultation” [6]. In 2018, Vogel et al. pointed out that video-assisted patient education can significantly improve patients’ level of knowledge and lead to increased patient satisfaction [7]. Furthermore, a position paper of the German Society for Anaesthesiology and Intensive Care Medicine (DGAI) and the German Society for Telemedicine (DG Telemed) from 2019 also called for using digital media to improve patient care and actively shape telemedicine in the field of anaesthesiology in Germany [3]. Given the safety of remote education and informed consent, it is plausible that both the risk and potential anxiety and discomfort for the child caused by a hospital visit are vastly reduced. However, the use of technology in this context was highly restricted prior to the COVID-19 pandemic [8,9].

One way to reduce the number and length of hospital stays, especially for children who have to undergo repeated procedures and examinations under anaesthesia, is patient informed consent using telemedicine. Benefits, such as cost savings, reduced waiting times, and increased patient safety, may be realized, especially for the vulnerable paediatric patient population. While the technology for digital support in patient informed consent has been possible for years, it remains unclear how widespread these opportunities actually are in clinical practice.

We therefore aimed to examine the current state of telemedicine’s use to obtain informed consent in anaesthesia in Europe. In particular, we aimed to investigate how anaesthetists judge the legal situation with regard to purely telemedical patient informed consent, the necessity of informing both parents in paediatric anaesthesia, and the necessity of parental/caretaker identity verification. In addition, the study examined whether there are different ways of obtaining informed consent depending on the severity of the procedure or surgery and whether the technical requirements are fulfilled to do so.

## 2. Materials and Methods

### 2.1. Study Protocol

To evaluate the recent practice of the informed consent process, a Europe-wide survey was created by the Department of Anaesthesiology at the University Hospital Bonn and conducted by ESAIC. The survey was sent to anaesthetists who consented to receive ESAIC informational emails in 47 European countries.

The survey focused on the use of digital support in the context of anaesthesia in general with a total of 27 questions, with a special focus on the peculiarities of patient information in paediatric anaesthesia. This was examined in six specific questions with three answer options each (for a detailed description see the Appendix A).

In addition to the collection of demographic data, such as gender or level of education, the anaesthetists were asked to express their opinions regarding the advantages and disadvantages of telemedicine-supported informed consent and whether telemedicine is at all possible in terms of the anaesthetist’s technical equipment. We included all survey answers (as opposed to only those that referred to paediatric anaesthesia) to illustrate the general use of digital media and their assessment as well as the potential for future use following the pandemic.

We used “LimeSurvey CE”, Hamburg, Germany (Version 5.1, https://community.limesurvey.org/downloads/, accessed on 29 November 2021) for the online questionnaire, hosted on a secured Linux Debian (Version 10.11) server. The survey was conducted over a three-week period (July to August 2021) and was supported by the ESAIC. An email containing a link to the survey was distributed to 42,433 active members by the ESAIC communication committee (https://kai-survey.de/limesurvey/733779/, accessed on 29 November 2021). According to the applicable medical professional regulations, an ethical approval was not necessary for this anonymous survey.

### 2.2. Questionnaire

Following previous projects [10,11,12,13], our working group developed an online questionnaire to analyse the use of telemedical support in healthcare and also specifically in paediatric anaesthesia in Europe. The questionnaire consisted of 27 mainly multiple-choice questions. Respondents were allowed to omit questions. The questionnaire could be completed in 7–10 min. Questions were asked concerning general anaesthesia issues, and a special part of the questionnaire was designed to consider paediatric anaesthesia. The general questions dealt with whether online informed consent is implementable in daily routine and whether anaesthesiologists know whether it is legal to obtain informed consent online depending on the country. Six questions were included in the special paediatric part of the questionnaire. The content of these questions was whether consent had to be obtained from both parents or only from one parent. A distinction was made here between simple procedures and high-risk procedures. In the next question, it was assessed whether the anaesthesiologist knew the legal regulations in his or her country for online informed consent. We asked whether the identity of the parents should be checked and if it is actually checked in routine practice. Furthermore, we asked the anaesthesiologists about the risks and benefits of online informed consent in paediatric anaesthesia.

We also investigated the general sentiment among participants of how informed consent should be obtained in the future and if ensuring certain guidelines might help to facilitate remote interviews in both adult and paediatric populations. Our survey thus assessed whether special regulations favouring informed consent online or per telephone in the general patient population were implemented during the pandemic situation.

### 2.3. Statistical Analysis

Data were summarized using descriptive statistics. For each question, the absolute number and the percentage of responses was calculated and used to interpret the opinion distribution. A chi-square test of independence was used for categorical variables. All analyses were performed using the programming language R (R Core Team, Vienna, Austria, Version 3.6.2). The threshold for statistical significance was set to *p* ≤ 0.05. To facilitate reproducible research, we programmed an interactive web application to investigate the data (https://kai-survey.shinyapps.io/ESAIC-KAI-survey-2021/, accessed on 29 November 2021). The code to perform descriptive statistical analysis and visualisation was stored at GitHub and can be viewed online (https://github.com/GrigorijSchleifer/ESAIC-informed-consent-survey/, accessed on 29 November 2021).

## 3. Results

### Population

The study population was defined as practicing ESAIC members. Respondents who did not provide their residency or submitted entries with missing values were excluded, leaving 930 eligible responses. Overall, responses were provided by medical doctors (99%, *n* = 920), nurses (0.2%, *n* = 2), physician assistants (0.6%, *n* = 6), and other undisclosed professions (0.1%, *n* = 1). Fifty-six percent (*n* = 521) of participants were male, 43.6% (*n* = 406) female, and 0.3% (*n* = 3) diverse. Respondents were predominantly consultants (78.6%, *n* = 731) or residents (15.3%, *n* = 143) (Table 1) from 47 countries (Appendix A). The highest number of responses came from Germany (*n* = 132), Spain (*n* = 73), and Switzerland (*n* = 65) (Figure 1).

We assessed the availability of technological solutions for the informed consent process across Europe. In 70.2% (*n* = 486) of the responses, we observed that it was not possible to obtain informed consent via the Internet in a routine setting. While 6.6% (*n* = 46) of those surveyed mentioned varying technical standards from day to day, 23.1% (*n* = 160) confirmed that it was possible to obtain consent via the Internet (Figure 2). Major differences could be observed for the technical distribution of tools supporting remote education across Europe (Figure 3).

The survey also investigated whether a remote consent process for paediatric patients in Europe was presumed to be legal. In paediatric practice, 67.3% (*n* = 432) of responses stated that online/telephone-based consent did not comply with legal regulations. Approximately a fifth (22.9%, *n* = 147) of respondents answered that online consent was legal for paediatric patients (Figure 4). Interestingly, there were major differences across European countries in perceived legal requirements for remote informed consent (Figure 5).

The answers to the question of whether written informed consent is required from both parents for elective surgery varied. For simple procedures, 14.2% (*n* = 92) of participants wished to obtain consent from both parents. In comparison, 25.7% (*n* = 167) of the responses stated that consent had to be obtained from both parents for complex procedures. For 77.6% (*n* = 502) of colleagues surveyed, informed consent for simple surgery could be given with only one parent present versus 63.8% (*n* = 415) for complex interventions (Figure 6).

We evaluated how identity of the legal representatives was confirmed. Overall, 56.1% (*n* = 362) of survey participants reported that it was necessary to see parents’ ID cards to confirm parental identity. However, only 29.9% (*n* = 193) of those surveyed actually verified identity by routinely checking ID when parents were present. There is an obvious discrepancy between the expectation of identification of legal representatives for paediatric patients and the routine practice (Figure 7).

Within our survey, we additionally asked the participants what concerns they had regarding telephone or online interviews. We found that major concerns were the lack of interaction, personal observation of the patient, and doctor–patient relationship. Overall, 26.8% (*n* = 632) said they were concerned about the lack of contact, 21.4% (*n* = 504) worried about missing out on personal observation of the patient, and 16.8% (*n* = 264) about a missing doctor–patient relationship. Legal uncertainty was also mentioned as a concern (Appendix A). We also assessed the general sentiment of the survey population toward the implementation of online/telephone interviews. Overall, 35.3% (*n* = 470) of respondents confirmed that a remote consent procedure could limit time spent waiting for the interview and could be more efficient than a face-to-face interaction (24.6%, *n* = 327). Furthermore, 23.1% (*n* = 308) of the study participants replied that remotely informed consent could be less stressful. From an organisational perspective, the use of standardized questionnaires was seen as an advantage by 16.8% (*n* = 224) of colleagues surveyed (Appendix A).

We also investigated the general sentiment among participants of how informed consent should be obtained in the future in both adult and paediatric populations. Overall, the majority (77.9% *n* = 589) of participants would prefer to do anaesthesiological preoperative evaluations in person. Only 7.5% (*n* = 57) could imagine obtaining consent online via a patient’s self-assessment or in a videoconference (11.6%, *n* = 88). Three percent (*n* = 22) favoured obtaining informed consent via telephone (Figure 8A). Interestingly, 60.2% (*n* = 414) confirmed that an online interview would be an alternative to personal face-to-face meeting if certain conditions were met.

Our survey assessed whether special regulations favouring informed consent online or per telephone in the general patient population were implemented during the pandemic. While 25.2% (*n* = 202) of participants did not know of any regulations, 56.1% (*n* = 450) of responses denied any regulatory adjustments favouring online consent. Only 18.7% (*n* = 150) reported that regulations favouring informed consent either online or per telephone were implemented in their countries (Figure 9).

## 4. Discussion

Our survey showed, for one, that the implementation of telemedicine in daily practice has been very limited across Europe and, for another, that perceived legal bases are very different among the individual countries.

In our questionnaire, 70.2% of responders stated that in their hospitals, it would not be possible to provide patient education via the Internet or telephone in a routine setting. In the United States, a collective of healthcare academics, researchers, providers, and industry representatives has called for healthcare reform coupled with telemedicine and information technology for many years [14]. Closing the gaps between technical feasibility and usability in everyday clinical practice remains a challenge.

In addition to the technical and practical problems, there is also the legal aspect of anaesthesiological information. Specifically relating to paediatric anaesthesia, 67.3% of respondents stated that purely digital education without any in-person interaction is not legal. Only 22.9% considered online or telephone-based education to be legal, while 9.8% were unable to assess the current legal situation. This was also extremely different among the participating European countries and underpins the fact that there is still no uniform regulation in this regard (Figure 5) and that even within a country, there is still great uncertainty concerning legality. The legal basis is quite different throughout Europe; in some countries, online or telephone education of patients is legal, whereas in others, only face-to-face education is [15,16].

European legislation considers telemedicine to be a health service on the one hand and an information service on the other and therefore is subject to different legislation [8]. Specific legal regulations for the use and handling of telemedicine are lacking in many countries, and harmonization across the EU is often described as unfeasible, for one, because of data protection problems [8].

The European Society of Anaesthesiology and Intensive Care (ESAIC) has published a guideline on preoperative anaesthesiological information [17], with recommendations for preoperative evaluation and the benefits of digital media, both of which are defined in this manuscript.

Considering the aspect of parental presence, it became apparent that most anaesthesiologists considered consent from only one parent sufficient. However, this also differed with regard to the complexity of the intervention (77.6% versus 63.8% for simple versus complex procedures, respectively). On the other hand, 14.2% of anaesthesiologists (for simple procedures) and 25.7% (for complex procedures) stated that consent of both parents was required. This may imply that both parents have to take time off work for the consultation, which can of course have economic or financial consequences and is not necessarily perceived as comfortable. Added to this are the sometimes not inconsiderable waiting times in the preoperative outpatient departments. This could mean an advantage for parents using online or telephone-based education.

A special question was designed to address the verification of parental identity. Although 56.1% of anaesthetists found it necessary, for example, to be shown a parent’s ID card, only 29.9% of the respondents said they actually practised this. Here, too, telemedicine could standardise the procedure: a QR code could be used to send a scan of the ID card to the doctor, for example, and many countries already have electronic ID cards that can be scanned online.

Especially during the COVID-19 pandemic, minimising personal contacts is an essential preventive factor to avoid further spread of the virus [17,18]. Children in particular are vulnerable and must be protected. This raises the question of whether the use of telemedicine can replace the personal patient presentation and whether there are adverse effects for patients to be feared as a result.

As an outlook for the future, it remains to be noted that in the overall group of respondents, only 22% would favour patient education online or via telephone. However, by differentiating between adult and paediatric anaesthesia, we found that 60.2% see digital education as an alternative to the face-to-face educational interview given optimal organisation of the process. This includes ensuring a specific guideline, which may include, e.g., a specific appointment time for the interview, a secure data-protected online service, a relaxed environment without distractions, and explicit parental consent for an online or telephone interview.

### Limitations

Although the questionnaire was sent to 42,433 active members of the ESAIC, only 930 anaesthetists took up the offer to participate in the survey. This corresponds to 2.19% of the respondents, and therefore, the data presented are limited in their representativity.

Nevertheless, an interesting picture emerges of how different the use of digital media in medicine is among European countries.

In countries with limited resources, it may be more difficult to implement the use of online tools, as patients may not have access to computers or the internet. Thus, at least in the near future, a widespread use of telemedicine to support health care in all European countries is probably not possible although it is desirable. One further limitation of the survey is the possibility of a response bias. Because the survey was anonymous and performed online without any human interaction, socially desirable responses and interviewer bias should not be of concern. Even though our questionnaire was designed with maximum care, a pre-test to evaluate a possible bias effect of primer questions could not be included because the survey was distributed by the ESAIC only once. This could potentially induce question-order effects that would limit the external validity of our findings.

## 5. Conclusions

According to international studies, digital patient education has proven to be feasible, safe, efficient, and cost-effective and has been increasingly recommended and used during the COVID-19 pandemic [6,8]. This paper highlights the current standard in Europe and filters out concerns anaesthetists have about a telemedicine approach while pointing out the potential benefits. Overall, the approach in Europe is extremely varied, not least because of varying legislation in the individual countries.

It would be desirable to create a uniform legal basis, especially for digital online anaesthesiological patient informed consent, to minimise unnecessary personal contacts while maintaining the same standard of information—even beyond a pandemic situation. Consequently, there would be greater opportunity to protect the health of particularly vulnerable and severely ill groups, such as those of children with congenital heart defects.

## Figures and Tables

**Figure 1 jcm-11-00502-f001:**
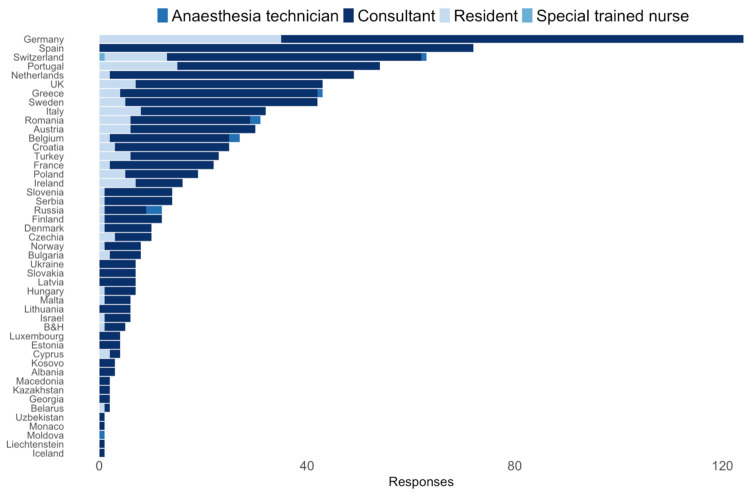
Responders’ level of expertise separated by the country.

**Figure 2 jcm-11-00502-f002:**
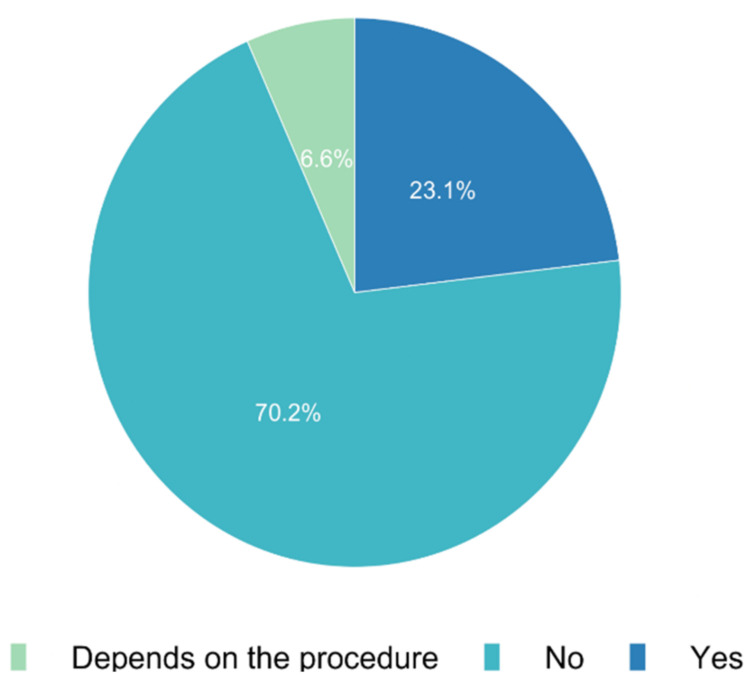
Is it possible to obtain informed consent online via the Internet in your routine setting?

**Figure 3 jcm-11-00502-f003:**
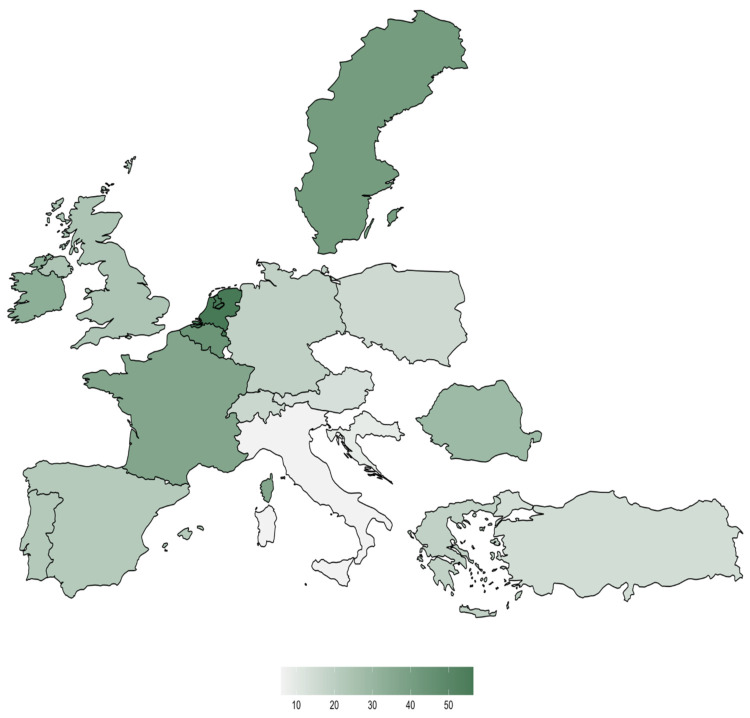
Is it possible to obtain informed consent online via the Internet in your routine setting? Only countries with at least 15 responses were colour coded.

**Figure 4 jcm-11-00502-f004:**
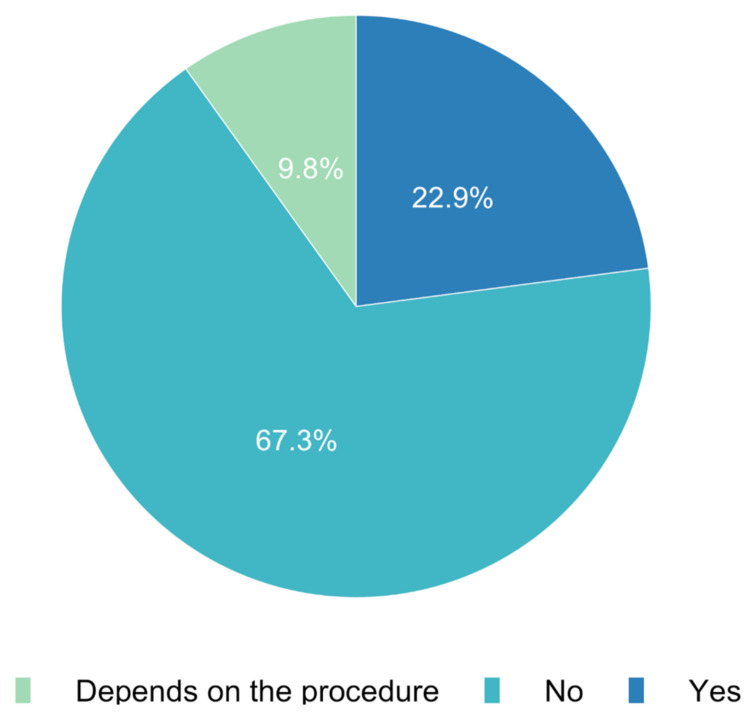
Do you know if it is legal to obtain informed consent from the parent/caregiver via the Internet or telephone?

**Figure 5 jcm-11-00502-f005:**
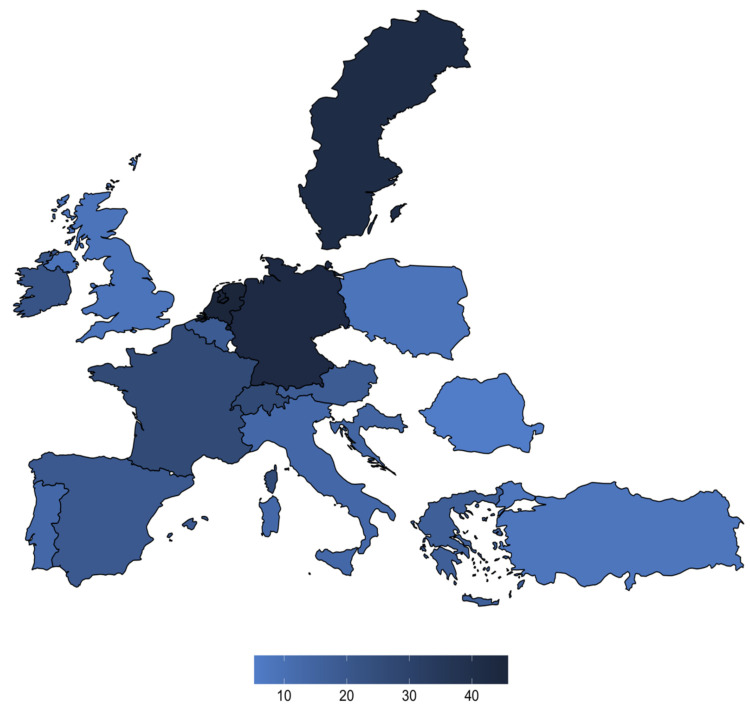
Do you know if it is legal to obtain informed consent from the parent via the Internet or telephone? Only countries with at least 15 responses were colour coded.

**Figure 6 jcm-11-00502-f006:**
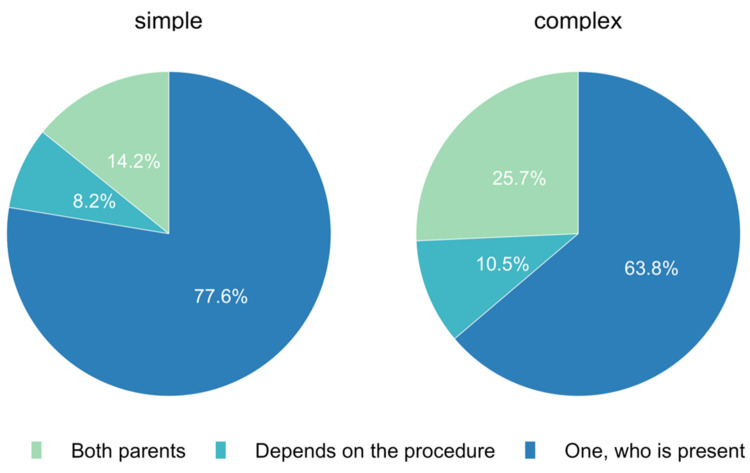
Is written informed consent for elective surgery required from both parents or just from one?

**Figure 7 jcm-11-00502-f007:**
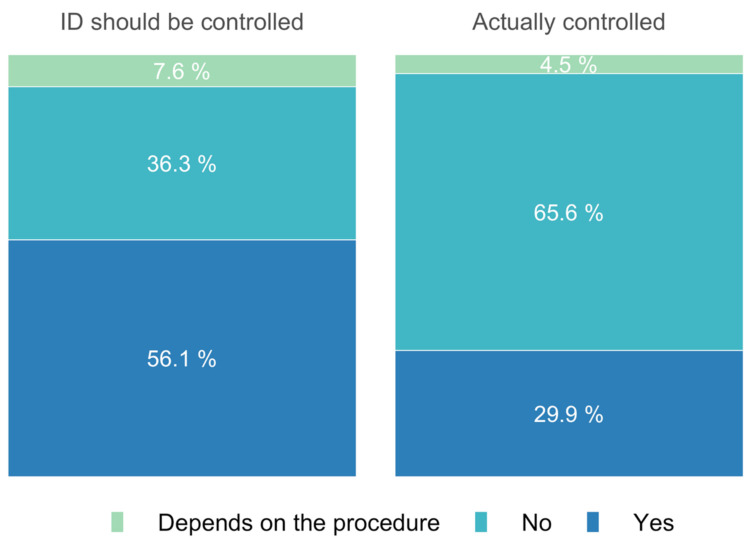
Do you think it is necessary to verify the identity of the parent/guardian (i.e., to check their ID card)?

**Figure 8 jcm-11-00502-f008:**
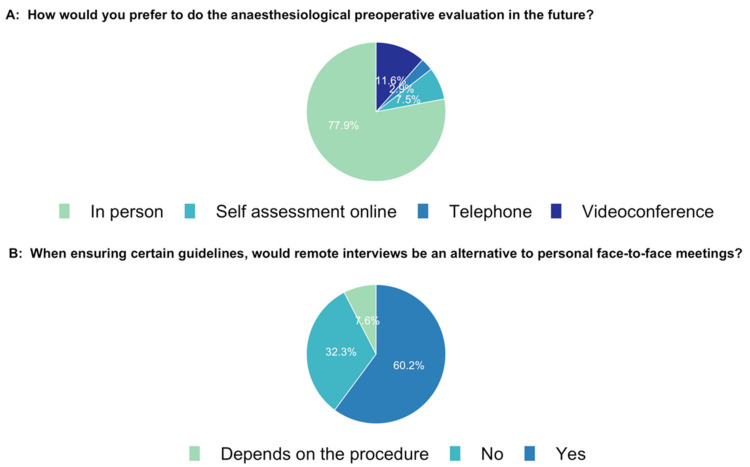
How would you prefer to conduct anaesthesia preoperative evaluations in the future?

**Figure 9 jcm-11-00502-f009:**
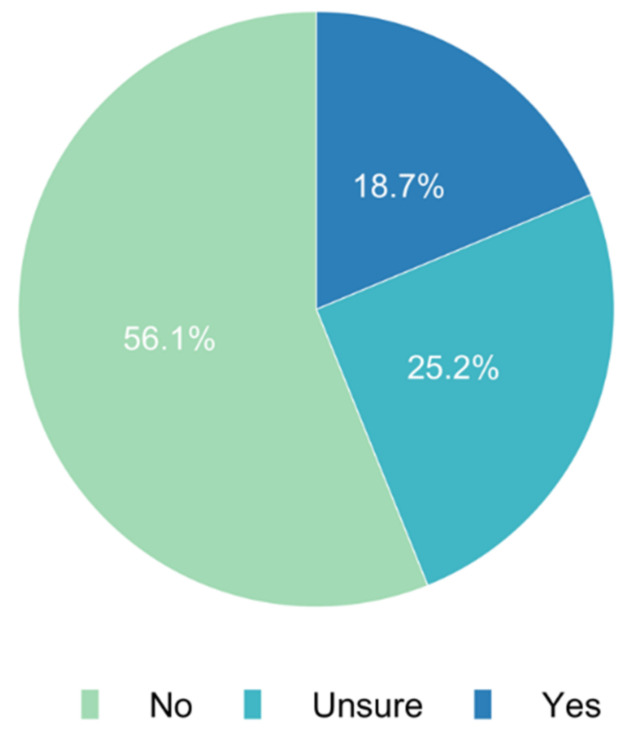
Do special regulations exist in your country during the current pandemic situation favouring online or telephone-based informed consent?

**Table 1 jcm-11-00502-t001:** Distribution among medical professions and level of expertise of the study population stratified by gender.

	Stratified by Gender			
	Diverse	Female	Male	*p*
*n* (%)	3 (0.3)	406 (43.6)	521 (56)	
**Profession (%)**				<0.001
Medical doctor	2 (66.7)	402 (99.0)	516 (99.2)	
Nurse	1 (33.3)	0 (0.0)	1 (0.2)	
Physician assistant	0 (0.0)	4 (1.0)	2 (0.4)	
Other	0 (0.0)	0 (0.0)	1 (0.2)	
**Expertise (%)**				0.33
Anaesthesia technician	0 (0.0)	4 (1.0)	6 (1.2)	
Consultant	2 (66.7)	303 (74.8)	426 (81.8)	
Resident	1 (33.3)	75 (18.5)	67 (12.9)	
Special trained nurse	0 (0.0)	0 (0.0)	1 (0.2)	
Other	0 (0.0)	23 (5.7)	21 (4.0)	

## Data Availability

Complete dataset is available on https://kai-survey.shinyapps.io/ESAIC-KAI-survey-2021/ (accessed on 29 November 2021). The code to perform descriptive statistical analysis and visualisation was stored at GitHub and can be viewed online https://github.com/GrigorijSchleifer/ESAIC-informed-consent-survey/ (accessed on 29 November 2021).

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
