# Peer review of "Digital Online Anaesthesia Patient Informed Consent before Elective Diagnostic Procedures or Surgery: Recent Practice in Children—An Exploratory ESAIC Survey (2021)"

_jcm, 2022, doi:10.3390/jcm11030502_

Round 1

Reviewer 1 Report

In this study authors aimed to examine the current state of use of tele medicine to obtain informed consent in anaesthesia in Europe and in particular, to investigate how anaesthetists judge the legal situation with regard to purely tele medical patient informed consent the necessity of informing both parents in paediatric anaesthesia, and the neces- sity of parental/caretaker identity verification. This is very interesting well-designed study.

Abstract section: please refers to the aim of the study.

The number of  recipients /  responses(42433/930), only 2,19% were responded, could mentioned, discussed or referred as limitation in the study

Author Response

We are very pleased about your positive evaluation of our study.

  1. Abstract section: please refers to the aim of the study.

We thank the reviewer for this remark. A sentence to this effect has been added to the abstract (line 35-36).

  1. The number of  recipients /  responses(42433/930), only 2,19% were responded, could mentioned, discussed or referred as limitation in the study

Thank you very much for this comment. Unfortunately, the turnout was not as high as we expected. We have added another subsection in the manuscript (4.1) and point out the limitations of our study. (lines 318-323)

Reviewer 2 Report

JCM_review                                                                                                      Dec 9, 2021

This manuscript is describing a survey from the ESAIC regarding online informed consent, and is potentially a topic of acute and timely relevance, however I have several concerns:

-the manuscript could have benefited from further review and editing, (e.g. telemedicine line 90 vs. tele medicine line 95); similarly for corona pandemic line 33 vs. COVID-19 pandemic line 59.

-formatting of the author affiliations, why not all listed x1 with unique e-mails separate?

-line 38, pediatrics should be plural

-line 40, setting and should read as two separate words.

-please re-read the intro. there are sections that are repetitive in nature.

-Materials and Methods:  -were individuals required to provide consent through the “ESAIC newsletter”, this was not clear or stated at all.

-how were the 27 questions generated, was there a survey scientist involved, was this validated?  How did you account for response bias (i.e. priming questions, that can provoke a certain response?) where appropriate expertise/resources obtained?  There are entire books written on this subject: https://oxford.universitypressscholarship.com/view/10.1093/acprof:oso/9780199747047.001.0001/acprof-9780199747047

-this is key and greater details should have been included in the methods.

-how does this subject presented apply to those patients in limited resource settings, w/ out internet (i.e. marginalized, underserved, under resourced), this online tool online applies to those with computers and internet.  This should be added to the discussion.

-figures should be legible in black and white print.

-the vast majority of the responses re. figure 2 & 3 where that informed consent was not possible via the internet in routine setting (70%), nor did practitioners think that it was legal (67%), however the title of the article provides a different lens on the work.  It might be encouraged to include “exploratory” or “preliminary” in the title.

-this work raises other glaring concerns which surrounds Fig 7.  This varying difference is challenging to digest, i.e. although almost 60% agree that ID should be controlled, only half that report as being done!  This survey demonstrates incidental findings, which again, perhaps could have been reviewed with appropriate survey science.  Was this intentional?  What might be the consequences of this?  False consenting?

-Lines 228-231 are contradictory to the intent of the manuscript, i.e. that the vast majority of the responses indicate that given the choice they would prefer to consent in person.  This is challenging and again is there some reporting bias as play.  How were the 27 questions amassed & designed?  On the one hand you are exploring an assessment of provider knowledge, and on the other had you are also reviewing preferences…it is a bit confusing.

-Discussion:  you are delving into legislation; would this be more helpful for the reader having this information at the beginning of the article?  Especially for North American or Asian readership, whom may not be familiar with the European landscape.

-there is an entire section missing at the end of the articles, funding, author contributions, consenting process, IRB approval, this should be added in complete detail.

-references are sparse, and there are no self-citations, which could be interpreted as the authors having no prior experience or expertise in the field.

-supplemental, there are different fonts used in Table 3.

I apologize for not being able to have a more favorable review, however the article needs more work.

Author Response

We thank the reviewer for the detailed comments and remarks.

  1. The manuscript could have benefited from further review and editing, (e.g. telemedicine line 90 vs. telemedicine line 95); similarly for corona pandemic line 33 vs. COVID-19 pandemic line 59.

Following the reviewer’s comment, care has now been taken to use consistent spelling throughout the manuscript. A native speaker did read the manuscript once again carefully. The changes are highlighted in track changes mode.

  1. Formatting of the author affiliations, why not all listed x1 with unique e-mails separate?

As suggested by the reviewer, the author details and e-mail addresses have been changed accordingly.

  1. line 38, pediatrics should be plural
  2. line 40, setting and should read as two separate words.

Some spelling mistakes have been corrected, the changes are highlighted in the track changes mode.

  1. Please re-read the intro. there are sections that are repetitive in nature

We deleted repetitive sentences (lines 61-62, 88, 89)

  1. Materials and Methods:  -were individuals required to provide consent through the “ESAIC newsletter”, this was not clear or stated at all.

ESAIC selects one survey each month created by its members on a wide range of medical topics to send out. For this purpose, active members receive corresponding links, which they can select if they are willing to participate in an online survey. The consent to receive these surveys is given in the general communication policy of the ESAIC. Participation is voluntary and completely anonymous. To make this clearer, a corresponding sentence has been added to the Material and Methods section (lines 109-110).

  1. How were the 27 questions generated, was there a survey scientist involved, was this validated?  How did you account for response bias (i.e. priming questions, that can provoke a certain response?) where appropriate expertise/resources obtained?  There are entire books written on this subject: https://oxford.universitypressscholarship.com/view/10.1093/acprof:oso/9780199747047.001.0001/acprof-9780199747047

We did not include a survey scientist, because we did not want to generate a validated questionnaire to produce reproductive results for years to come. Our aim was to give an overview of the current situation in Europe. We did add a statement in the new limitations section (line 327-333).

  1. This is key and greater details should have been included in the methods.

As we did not validate our questionnaire, we stated the following in the limitations: “One further limitation of the survey is the possibility of a response bias. Because the survey was anonymous and was performed online without any human interaction, socially desirable responding and interviewer bias should not be of concern. Even though our questionnaire was designed with maximum care, a pretest to evaluate a possible bias effect of primer questions could not be included because the survey was distributed by the ESAIC only once. This could potentially induce question-order effects that would limit the external validity of our findings.” (lines 327-333)

  1. How does this subject presented apply to those patients in limited resource settings, w/ out internet (i.e. marginalized, underserved, under resourced), this online tool online applies to those with computers and internet. This should be added to the discussion.

The reviewer raises a very interesting point. We have tried to show the inequal distribution of digital support and technology as an overview with the country map (Figure 2), as it is not equally represented in all countries. We have added a corresponding explanatory paragraph: “Furthermore, in countries with limited resources, it may be more difficult to implement the use of online tools as patients may not have computers or internet access. Thus, at least in the near future, the widespread use of telemedicine to support health care in all European countries is probably not possible, although desirable.” (lines 324-325)

  1. Figures should be legible in black and white print.

As requested, the colours in the illustrations have been adjusted so that they are also legible when the article is printed in black and white.

  1. The vast majority of the responses re. figure 2 & 3 where that informed consent was not possible via the internet in routine setting (70%), nor did practitioners think that it was legal (67%), however the title of the article provides a different lens on the work.  It might be encouraged to include “exploratory” or “preliminary” in the title.

We appreciate this valuable comment. In fact, our Europe-wide survey showed that the use of telemedicine support in routine settings is not yet widely established. Similarly, there is often confusion about the legality of telemedicine use due to differences in legislation between countries. All of this came into sharper focus in the context of the COVID-19 pandemic, as the advantage of remote education corresponds to the desire for contact restrictions.

Thus, the word "exploratory" has been added to the title to clarify that the survey does not fulfil the criteria of a representative questionnaire (line 4).

  1. This work raises other glaring concerns which surrounds Fig 7.  This varying difference is challenging to digest, i.e. although almost 60% agree that ID should be controlled, only half that report as being done!  This survey demonstrates incidental findings, which again, perhaps could have been reviewed with appropriate survey science.  Was this intentional?  What might be the consequences of this?  False consenting?

Thank you very much for this critical remark. The questions were based on aspects that the participants encounter in the course of their clinical work. For example, whether parents have to identify themselves is not uniformly handled within Germany or even within a hospital. The specific question of identity, as well as the question of if and when both parents must consent, should represent actual practice in the context of this descriptive survey. Nevertheless, this result was also surprising for the authors. We regret that no further questions were asked about the reasons; this was also due to the brevity of the questionnaire, which was supposed to be completed in a maximum of 10 minutes. Nevertheless, in the opinion of the authors, the available data are worth publishing.

  1. Lines 228-231 are contradictory to the intent of the manuscript, i.e. that the vast majority of the responses indicate that given the choice they would prefer to consent in person.  This is challenging and again is there some reporting bias as play.  How were the 27 questions amassed & designed?  On the one hand you are exploring an assessment of provider knowledge, and on the other had you are also reviewing preferences…it is a bit confusing.

The intent of the manuscript is not to favour or to decline online informed consent. We just asked the participants about the current practice in their country (figure 2 to 5), what they would personally prefer in the future (figure 8 A), and if they thought that it might become possible to perform remote interviews in the future (figure 8 B). We designed the questions in a team with experts for paediatric anaesthesia, clinical trials, and legislation.

  1. Discussion:  you are delving into legislation; would this be more helpful for the reader having this information at the beginning of the article?  Especially for North American or Asian readership, whom may not be familiar with the European landscape.

The main focus of this descriptive study was on the implementation of telemedical support in daily practice and the anaesthetists' subjective assessment of how they implement it in accordance with the law. Therefore, the discussion only briefly touched on the fact that European countries handle it very differently. A more detailed description of the legal situation in each individual country would eclipse  the scope of this publication.

  1. There is an entire section missing at the end of the articles, funding, author contributions, consenting process, IRB approval, this should be added in complete detail.

Some of this information (funding, author contribution) was requested separately when the paper was uploaded and was obviously unfortunately not available to the reviewer. The relevant section including the requested information about ethics has now been added to the methods section of the article. (lines 127-128)

  1. References are sparse, and there are no self-citations, which could be interpreted as the authors having no prior experience or expertise in the field.

Thank you for your insight. We understand your remark as a motivation to highlight that most of the authors have been working together for more than 50 professional years in research and science. Furthermore, some of the authors are regularly involved in ESAIC pan-European studies and even serve on steering committees. Thus, there is a great deal of expertise in the development and implementation of clinical trials. In addition to the accurate creation of eCRF's, this also includes the implementation of clinically relevant questions in corresponding surveys, also in multicentre projects. To support this point, a corresponding sentence has been added under point 2.2 and reference has been made to corresponding self-citations.

  1. Supplemental, there are different fonts used in Table 3.

The font in Table 3 was adapted to the rest of the text.